# A Pictorial Exploration of Mammary Paget Disease: Insights and Perspectives

**DOI:** 10.3390/cancers15215276

**Published:** 2023-11-03

**Authors:** Luciano Mariano, Luca Nicosia, Davide Pupo, Antonia Maria Olivieri, Sofia Scolari, Filippo Pesapane, Antuono Latronico, Anna Carla Bozzini, Nicola Fusco, Marta Cruz Blanco, Giovanni Mazzarol, Giovanni Corso, Viviana Enrica Galimberti, Massimo Venturini, Maria Pizzamiglio, Enrico Cassano

**Affiliations:** 1Breast Imaging Division, AOU Città della Scienza e della Salute di Torino, 10126 Turin, Italy; luciano.mariano@ieo.it; 2Department of Biotechnology and Life Sciences, University of Insubria, Via J.H. Dunant, 3, 21100 Varese, Italy; 3Breast Imaging Division, IEO—European Institute of Oncology IRCCS, Via Ripamonti, 435, 20141 Milan, Italy; filippo.pesapane@ieo.it (F.P.); antuono.latronico@ieo.it (A.L.); annacarla.bozzini@ieo.it (A.C.B.); maria.pizzamiglio@ieo.it (M.P.); enrico.cassano@ieo.it (E.C.); 4Radiology Division, Department of Precision Medicine, University of Campania Luigi Vanvitelli, 80138 Naples, Italy; davide.pupo@ieo.it; 5Department of Diagnostics and Public Health, University of Verona, Piazzale L.A. Scuro 10, 37134 Verona, Italy; antoniamaria.olivieri@studenti.univr.it; 6Postgraduation School in Radiodiagnostics, Faculty of Medicine and Surgery, University of Milan, 20122 Milan, Italy; sofia.scolari@unimi.it; 7Division of Pathology, IEO, European Institute of Oncology IRCCS, 20141 Milan, Italy; nicola.fusco@ieo.it (N.F.); marta.cruzblanco@ieo.it (M.C.B.); giovanni.mazzarol@ieo.it (G.M.); 8Department of Oncology and Hemato-Oncology, University of Milan, 20122 Milan, Italy; giovanni.corso@ieo.it; 9Division of Breast Surgery, European Institute of Oncology (IEO), IRCCS, Via Ripamonti, 435, 20141 Milan, Italy; viviana.galimberti@ieo.it; 10European Cancer Prevention Organization (ECP), 20122 Milan, Italy; 11Diagnostic and Interventional Radiology Unit, ASST Settelaghi, Insubria University, 21100 Varese, Italy; massimo.venturini@uninsubria.it

**Keywords:** paget’s disease mammary, breast neoplasms, prognosis

## Abstract

**Simple Summary:**

This review provides a comprehensive overview of mammary Paget disease (MPD) from its historical origins to the current diagnostic and treatment strategies while also exploring promising avenues for future advancements in the field. With improved understanding and ongoing research efforts, we aim to enhance the prognosis and quality of life for individuals affected by this relatively uncommon condition.

**Abstract:**

Mammary Paget disease (MPD) is a rare condition primarily affecting adult women, characterized by unilateral skin changes in the nipple–areolar complex (NAC) and frequently associated with underlying breast carcinoma. Histologically, MPD is identified by large intraepidermal epithelial cells (Paget cells) with distinct characteristics. Immunohistochemical profiles aid in distinguishing MPD from other skin conditions. Clinical evaluation and imaging techniques, including magnetic resonance imaging (MRI), are recommended if MPD is suspected, although definitive diagnosis always requires histological examination. This review delves into the historical context, epidemiology, pathogenesis, clinical manifestations, and diagnosis of MPD, emphasizing the need for early detection. The classification of MPD based on pathogenesis is explored, shedding light on its varied presentations. Treatment options, including mastectomy and breast-conserving surgery, are discussed with clear guidelines for different scenarios. Adjuvant therapies are considered, particularly in cases with underlying breast cancer. Prognostic factors are outlined, underlining the importance of early intervention. Looking to the future, emerging techniques, like liquid biopsy, new immunohistochemical and molecular markers, and artificial intelligence-based image analysis, hold the potential to transform MPD diagnosis and treatment. These innovations offer hope for early detection and improved patient care, though validation through large-scale clinical trials is needed.

## 1. Introduction and Historical Pills

Mammary Paget disease (MPD) is a relatively uncommon condition affecting predominantly adult women, which is characterized by monoliteral skin changes in the nipple–areolar complex (NAC) and frequently associated with an underlying in situ (DCIS) or ductal invasive breast carcinoma (BC) [1,2]. The prevalence of an associated cancer ranges from 67% to 100% with most studies reporting a concurrent malignancy in over 90% of patients [1].

It was initially described by John of Arderne in 1307 [3] and by Velpeau later in 1856 as erythematous–eczematous changes in the NAC [4,5], but it was James Paget, two decades later, who reported a relationship between a nipple rash and mammary gland tumor that gave the name to the disease [6]. Although patients with a palpable mass and/or suspicious digital mammography (DM) and ultrasound (US) findings have a high probability of invasive cancer and worse outcomes [7], even in patients without radiological abnormalities but with clinical symptoms, there is the possibility of a pathology that may need treatment. In those cases, magnetic resonance imaging (MRI) is an effective diagnostic tool for detecting clinical and conventional imaging occult cancer through the evaluation of breast vascularization. Many reports have indicated the usefulness of breast MRI in evaluating patients with MPD [8,9,10]. This review provides an overview of the existing literature on MPD, encompassing its histopathologic features, pathogenesis, and clinical manifestations. The differential diagnosis of nipple changes and the classification of MPD will be discussed, and we will demonstrate the imaging findings in DM, US, and MRI. Additionally, we address the management of MPD and the role of these imaging techniques in treatment planning.

## 2. Epidemiology and Main Risk Factors

MPD represents approximately 1–3% of all BC cases [1,2] with a prevalence among postmenopausal women aged 50 to 60 years [11,12], although it can occur in both sexes at any age. Male patient MPD accounts for approximately 1% of cases with a mean diagnostic age of 68 years [13]. The incidence of MPD varies geographically with higher rates reported in Western countries, probably related to lifestyle. An intriguing observation is the gradual decline in MPD cases from the 2000s, potentially attributed to an increased detection of early DCIS cases through mammographic screenings [14]. Specific risk factors for MPD are not identified. The same risk conditions known for BC are also applicable to MPD, including advanced age, family history of BC, inherited genetic mutations (BRCA 1 and/or BRCA 2), hormonal exposure (as hormone-replacement therapy and prolonged use of oral contraceptives), obesity, alcohol consumption, and prior thoracic irradiation, especially in women of a young age with high dosages of exposure.

## 3. Pathogenesis and Classification

The exact etiology of MPD needs to be better understood. Several hypotheses have been proposed over the years to explain its pathogenetic mechanism and influence subsequent therapeutic approaches, including epidermotropic [ET] and transformation [TF] theories [15]. The most widely accepted is ET, which suggests an initial underlying DCIS or invasive BC with a subsequent migration of Paget’s cells (PC) from the malignancy to the epidermis of the NAC through the ductal system without crossing the basal membrane [16]. This theory finds support in the observation that, in over 90% of MPD, an underlying BC is recognized (instances where tumors are not detected involve lesions of a size that falls below the detection threshold of conventional imaging techniques) [17] as well as similarities of immunohistochemical profile and gene expression patterns between PC and the underlying tumor: PC’s migration is thought to be mediated by an interaction between human epidermal growth factor receptor 2 (HER2/neu) on PC and heregulin-alpha, a motility factor released from epidermal keratinocytes that is able to promote a chemotactic action. Indeed, several studies have demonstrated a relatively pronounced HER-2/neu overexpression on PC, 90%, compared to DCIS or invasive BC expression without MPD, 50% and 20%, respectively [18,19].

On the other hand, TF proposes a local malignant transformation of the NAC keratinocytes in PC, leading to the development of MPD [17,20] as an independent process compared to the underlying mammary parenchyma [15]. This concept finds additional support in the following aspects: MPD cases without underlying BC or with an underlying tumor unrelated to NAC disease; ultrastructural studies [21] have demonstrated microvilli and desmosomal connections between PC and keratinocytes, indicating an intraepidermal origin of PC; the identification of pre-PC (Toker’s cells), clear cytoplasm suprabasal cells with characteristics of both keratinocytes and PC [22,23]; different genetic patterns between PC and underlying carcinoma cells [24]. Based on those pathogenetic notions, MPD can be categorized into three groups [15,25]: (a) only NAC-MPD without DCIS associated; (b) NAC-MPD with underlying lactiferous ducts DCIS; and (c) NAC-MPD with underlying lactiferous ducts DCIS and DCIS or invasive BC elsewhere at least 2 cm from the NAC (Figure 1). TF theory could potentially explain subtype and is most uncommon; in one of the most extensive studies involving 34 patients with MPD, Morrogh et al. [26] found only two cases of MPD without DCIS. However, it more frequently represents cases where the underlying primary BC is not detected with current imaging techniques and is diagnosed a few years later.

## 4. Pathology

The histological hallmark of MPD is the presence of large intraepidermal epithelial cells (i.e., PC) with abundant pale cytoplasm, pleomorphic nuclei, and prominent nucleoli [27,28] (Figure 2A). PC can be found as solitary elements or in clusters dispersed throughout the epidermis and exhibit variable mitotic activity. The formation of glandular structures is infrequent. In 4–8% of cases, PC may extend into the dermis with their cytological characteristics remaining akin to those observed in the epidermis. Notably, there is a distinct demarcation between the infiltrating PC and the underlying BC with the possibility of a concomitant lymphocytic infiltrate in the interface [29]. To differentiate MPD from other malignant processes involving the skin (e.g., Bowen disease, inflammatory BC, melanoma in situ, or squamous cell carcinoma in situ), immunohistochemistry may be of help [30]. PC are typically positive for low-molecular-weight cytokeratins (e.g., CK7 and CAM5.2), but they can rarely be negative or only focally positive; however, it is important to note that pagetoid squamous cell carcinoma in situ can also express these markers. HER2 is overexpressed in 80–90% of cases (Figure 2B,C); ER and PgR are positive in approximately 40% and 30% of MPD cases, respectively. However, it is worth noting that PC usually share the same immune profile as the underlying carcinoma [24].

In a recent investigation, a novel immunohistochemical marker, TRPS1, was introduced and has since become widely adopted for confirming the diagnosis of MPD [31]. According to the findings reported in this study, TRPS1 expression was detected in 100% of cases of MPD. This breakthrough in immunohistochemistry has proven to be invaluable in distinguishing MPD from histological mimics, notably melanoma in situ and squamous cell carcinoma in situ. PC are negative for melanocytic markers (e.g., SOX10) (Figure 2D). Furthermore, it is worth noting that squamous cell carcinoma in situ is an exceedingly rare occurrence within the NAC. In such cases, the diffuse presence of TRPS1 within an intraepithelial pagetoid lesion arising in the nipple invariably confirms the diagnosis of MPD.

## 5. Clinical Presentation

MPD typically presents as unilateral NAC cutaneous changes, including nipple itching, erythema/eczematous lesions, erosion/ulceration, nipple retraction, and serous/bloody discharge, individually or in combination [1] (Figure 3). Lesions usually arise from the nipple and spread to involve the surrounding areola complex [32], but cases of perimammary skin and opposite breast disease extending have also been reported [33,34,35]. Pain, itching, and a burning sensation occur in 15–25% of cases and often precede physical manifestation [36,37]; ulceration, bleeding from the nipple, and serous discharge destruction may occur in advanced stages [38]. A palpable mass may or may not be revealed to clinical examination and is usually correlated with the patients’ outcomes [38]. The mass is typically centrally located, although it can also be situated on the periphery of the breast in up to 41% of cases, showing a multifocal or multicentric localization [5,11], according to the capacity of the PC to propagate via the ducts.

Three possible distinct clinical patterns have been identified: (1) NAC changes with palpable mass associated, (2) NAC changes without palpable mass associated, and (3) subclinical MPD incidentally discovered on a pathology specimen of a breast mass [38]. Multiple research studies validate that approximately 50% of patients diagnosed with MPD exhibit a detectable breast mass, and 90–94% are linked to underlying invasive disease. Conversely, instances involving a lesion that lacks a discernible mass typically correlate with DCIS [1,37,39,40]. Some patients may not exhibit suggestive signs of MPD with an occasional PC finding through histopathologic NAC examination during mastectomy. The correlation between cancer spread extension and MPD method detection has been investigated: individuals with cutaneous NAC alterations generally exhibit less widespread lesions than those with clinically asymptomatic conditions [41]. Clinical NAC changes, although suggestive for MPD, can pose challenges in diagnosis. Misdiagnosis and delayed diagnosis are very common due to the similarity of skin manifestations with other benign inflammatory conditions (such as eczematous dermatitis, psoriasis, allergic contact or irritant dermatitis, and lichen simplex) [42], mainly when associated at temporary resolution with or without corticosteroid applications. Furthermore, in patients who undergo conservative surgery for breast cancer, skin observations frequently could be linked to post-surgical or radiotherapy (RT) effects. Ashikari et al. reported a median delay in the diagnosis and treatment of MPD of 6–11 months compared with 1–2 months for ordinary ductal carcinoma [1]. Benign skin changes are usually bilateral and may be associated with atopic systemic symptoms and quick response to topical steroid therapy [43]. Otherwise, monoliteral NAC skin involves persistence for more than three weeks following steroid application; MPD should be suspected [44], and a biopsy is necessary.

## 6. Diagnosis

A careful and timely evaluation, including clinical examination with imaging techniques integration (as DM, US, and MRI), is always required in all patients with NAC skin changes to exclude MPD. However, a conclusive diagnosis invariably necessitates histological scrutiny. In addition to the typical clinical skin features, a thorough clinical breast examination is essential for detecting potential masses and axillary lymphadenopathy, which are indicators of underlying infiltrative pathology [45]. Specific imaging patterns have also been documented. A thorough mammographic assessment needs a digital system with an integrated tomosynthesis technique and magnification views of the NAC and the breast anterior third for detecting any underlying lesions and ruling out multifocal disease, as MPD has reported prevalence rates of 41% and 34% for multifocality and multicentricity, respectively [46]. Notable DM findings include thickening cutaneous or nipple and periareolar skin retraction [15]. Malignant microcalcifications along with a subareolar mass may be present, limited to the retroareolar region or elsewhere in the breast (Figure 4). However, DM sensitivity and specificity are limited: abnormal mammographic findings involving the NAC are not specific for MPD, especially in women with prior surgery or RT; in addition, DM may appear negative in 22–25% of patients, leading to a potential underestimation of disease extent [41,47].

US is a non-invasive, repeatable, and widely available technique. Its inclusion in the initial assessment could benefit and warrants consideration, mainly when DM yields negative results, especially in women with denser breast tissue. Similar to mammographic MPD findings, US alterations are nonspecific and include expected changes for BC treatments, including hypoechogenic masses, microcalcifications, ductal ectasia, NAC flattening, asymmetry, and thickening [26,47,48] (Figure 5). Additionally, US offers an option of immediate image-guided intervention and improves the diagnostic accuracy for axillary lymph node status. However, not all underlying BC can be identified through the US. MRI’s higher sensitivity in evaluating the retroareolar region provides crucial information for clinically evident MPD cases with occult findings in DM and US and for the preoperative assessment of disease extent in patients eligible for breast-conserving therapy [26,47]. In addition, thanks to contrast enhancement, it shows nipple involvement [49]. MRI findings include asymmetric and abnormal NAC enhancement patterns, sometimes associated with non-mass-like enhancement or suspicious masses elsewhere in the breast [26,47,50], even a distant site with no apparent anatomic connection, about possible MPD multifocality and multicentricity (Figure 6). MRI may also evaluate lymph node status, raising concerns about axillary or internal mammary involvement. However, it is essential to note that false-negative MRI results for NAC evaluation in MPD cases have also been reported [25,50], probably for less-aggressive disease forms. Therefore, all clinically suspicious findings must undergo biopsy, regardless of negative imaging results. A full-thickness NAC biopsy with histological and immunohistochemical evaluation is the gold standard for establishing an ultimate diagnosis [38]. Research examining the molecular markers in MPD cases has revealed an expression of HER-2/neu, cytokeratin 7 (CK7), mucin 1 (MUC1), and human milk fat globule, and positive staining with CAM 5.2 antibody [51,52]. Exfoliative cytology with PC demonstration may be helpful, but a negative result can occur; its use has been postulated as an easy screening test for eczematous skin changes to the nipple.

## 7. Notes of Therapy

Accurate assessment and appropriate management of patients with MPD require a multidisciplinary discussion conducted by an expert team [53].

After diagnosis with a NAC punch-biopsy, breast surgery remains the first approach for the treatment. Mastectomy plus axillary staging or breast-conserving surgery (BCS) with or without axillary staging are the two main treatments considered for MPD. The literature reports controversial results about loco-regional recurrence after mastectomy or BCS for MPD. A recent meta-analysis, including seven studies with 685 patients, reported a cumulative local recurrence rate of 5–7% among women undergoing mastectomy and approximately 13% among those treated with BCS. Compared to the BCS group, mastectomy showed significant differences in terms of local recurrence (OR = 0.38, 95% confidence interval 0.21–0.69; *p* = 0.001) [54].

Following this study, BCS is contra-indicated in MPD treatment. However, not all women were treated with RT after BCS, and there is no information about invasive components that could be associated with MPD and/or about margin status after surgery. The European Organization for Research and Treatment of Cancer trial (EORTC 10873), published in 2001, demonstrated clearly that BCS plus RT is a feasible approach for patients with MPD if a clear margin was achieved at excision [55]. We can accept that BCS with complete NAC excision is a safe alternative to mastectomy, providing clear surgical margins and adjuvant RT [5,56,57].

Since MPD frequently underlies invasive carcinoma, in accord with the MD Anderson Cancer Center consensus, we can distinguish two conditions for breast surgery treatments: (a) NAC biopsy positive and imaging negative (including bilateral DM, ipsilateral US of the nodal basins, and US evaluation with a focus on the nipple/retro-areolar breast) after further evaluation (including breast MRI and biopsy of MRI-detected lesion) and (b) NAC biopsy positive and imaging positive (including bilateral DM, ipsilateral US of the nodal basins, and US evaluation with a focus on the nipple/retro-areolar breast) after further evaluation (including core biopsy of the breast lesion and considering breast MRI if the patient desires BCS) [58].

In the case of “no breast lesion, no microcalcifications, and NAC Paget only”, according to the size of the breast, central wide excision without axillary staging or mastectomy including NAC with sentinel node biopsy is suggested following appropriate systemic adjuvant and/or radiation therapy, depending on the stage and pathology.

In the case of “DCIS and NAC Paget”, mastectomy including NAC with axillary staging or BCS with complete excision of the NAC and excision of the breast tumor followed by whole breast radiation are recommended. In the case of “Breast invasive and NAC Paget”, mastectomy including NAC with axillary staging or BCS with complete excision of the NAC and excision of the breast tumor followed by whole breast radiation are recommended, also based on the extent and stage of the underlying BC.

In both situations, appropriate systemic adjuvant and/or RT are required. For example, in the case of MPD without invasive components or with associated DCIS ER-positive, low dose tamoxifen at 5 mg once daily for three years is recommended [59].

To conclude, the use of adjuvant systemic therapies has not been supported by any evidence, unless in cases with underlying BC [44], where adjuvant RT is also indicated after conserving surgery [55], especially with positive axillary lymph nodes, tumor size > 5 cm, or positive margins. Conversely, tamoxifen in male patients is generally used as the standard adjuvant therapy, since most male BC are estrogen-receptor positive [60].

## 8. Prognosis and Follow-Up

MPD prognosis is related to invasive components, axillary lymph node metastasis, and palpable mass. Invasive cancers associated with MPD are more likely to be high-grade, estrogen- and progesterone-receptor negative, and human epidermal growth factor receptor 2 positive than those in patients with no associated MPD. Regarding lymph node staging, 48–69% of patients with a breast mass and MPD have been found to have lymph node involvement compared with 21–25% where no mass is found. Lymph node metastasis is an independent predictor of poor outcomes [56]. MPD prognosis in patients without a palpable mass is generally good with a survival rate of 5 years in 90–100% of cases; conversely, women with MPD and a palpable mass associated generally have an underlying invasive carcinoma and a 5-year survival rate of approximately 20–65% [61]. Additional prognostic factors include tumor size, grading, and comorbidities of the patients. Male MPD prognosis is worse with a 5-year survival rate of approximately 20–30% [62]; it has been hypothesized that this may be due to the small size of the male mammary gland with early local and axillary extension.

## 9. Future Perspectives

Potential advancements in managing patients with MPD offer hopeful opportunities for early diagnosis and increased treatment efficacy. Several emerging techniques are being investigated, including new specific immunohistochemical and molecular BC markers and the increasingly emerging integration of artificial intelligence (AI) and deep-learning (DL) algorithms for the diagnosis of diseases based on image analysis. LB is a less intrusive method for identifying biomarkers from bodily fluids, like blood, urine, sputum, and saliva [63,64]. This technique has demonstrated successful application to anticipate treatment response and oversee EGFR-mutated patients with lung cancer undergoing tyrosine kinase inhibitor therapy [63]. The FDA’s recent approval of the Thera screen PIK3CA RGQ for detecting PIK3CA mutations extracted from the plasma of patients with BC has enhanced the role of LB in their management [65]. However, larger-scale studies are required to establish their actual clinical utility in the treatment of these patients, potentially enabling the differentiation of various histological BC patterns, including MPD types.

Immunohistochemistry and molecular profiling studies offer a valuable approach to diagnosis and are necessary to differentiate MPD from other malignant skin NAC processes. A recent study focused on aberrant glypican-3 (GPC3) expression in samples of BC and its potential as a subtype-specific biomarker and potential therapeutic target of some cancer patterns [66]. Assessing GPC3 expression in a cohort of 230 patients with BC revealed that 7.5% exhibited this particular marker, especially in subtypes such as MPD, intraductal carcinoma, and mucinous carcinoma. GPC3 expression was detected in all cases of MPD as well as in 42.9% of intraductal carcinomas and 16.7% of mucinous carcinomas.

AI, specifically DL techniques employing convolutional neural networks (CNNs), have garnered significant attention within the medical field due to their transformative impact on disease management through image analysis. Currently, there are over 20 FDA-approved AI applications for breast imaging, yet their adoption and utilization vary widely and are generally low. While a significant portion of the published literature and available AI applications predominantly concentrate on DM cancer detection, the potential applications of AI in breast imaging extend beyond this, encompassing risk assessment, breast density quantification, workflow optimization, triage, quality assessment, response evaluation to neoadjuvant therapy, and image enhancement [67]. An AI model trained on a dataset of over 1 million images exhibited an excellent AUC of 0.895 for BC detection, outperforming individual radiologists. However, the highest performance was observed when radiologists and AI were combined in a hybrid model [68]. Mirai, a deep-learning-based mammography risk model, integrates DM features and clinical factors to predict BC risk within a 5-year timeframe, showing robust validation across diverse international datasets [69]. Breast density, an independent risk factor for BC, demonstrates a moderate association with cancer risk. Several fully automated DL algorithms now employ CNNs for precise breast density stratification [70]. In a proof-of-concept study, Skarping et al. [71] illustrated the effectiveness of a DL-based model that employed baseline digital mammograms to predict patient responses to neoadjuvant therapy, achieving an AUC of 0.71.

Given their proficiency in extracting image characteristics, CNNs represent a specialized category of DL algorithms frequently employed for examining BC images [72]. These emerging advancements have demonstrated their potential to elevate the precision of imaging methodologies, thereby enhancing the management of patients with BC. Wu et al. developed and verified a DL method with five different CNNs, named ResNet34, ResNet50, MobileNetV2, GoogleNet, and VGG16, in Asian extramammary PD pathological image screening to distinguish between PCs and normal cells [73]. The ResNet34 model achieved the highest accuracy of 95.5% when applied to pathological images captured at a × 40 magnification level. This performance suggests its potential utility in enhancing the proficiency and precision of pathologists, ultimately leading to possible improvements in the care of patients with MPD. Further large-scale clinical trials are necessary to validate these emerging techniques and assess their efficacy and cost-effectiveness.

## 10. Conclusions

In conclusion, this comprehensive review of MPD sheds light on the intricate aspects of this relatively uncommon condition. MPD is a sentinel sign, almost invariably indicating an underlying breast malignancy. Its historical evolution, epidemiology, pathogenesis, and clinical presentation have been examined, underscoring the need for early detection and accurate diagnosis. The classification of MPD based on pathogenesis offers valuable insights into its diverse manifestations. The histological features of PC coupled with their immunohistochemical profiles aid in distinguishing MPD from other skin conditions. Clinical evaluation and imaging techniques, including MRI, are recommended if MPD is suspected, although definitive diagnosis always requires a nipple histological examination. This review discusses the nuances of treatment, emphasizing the importance of a multidisciplinary approach. Surgical options, including mastectomy and breast-conserving surgery, are presented with clear guidelines for different scenarios. Adjuvant therapies are considered, especially in cases with underlying breast cancer. Prognostic factors and their impact on patient outcomes are highlighted, underscoring the importance of early intervention in managing MPD. In the future, promising developments, such as liquid biopsy, immunohistochemical and molecular markers, and artificial intelligence-based image analysis, can potentially revolutionize MPD diagnosis and treatment. These emerging techniques offer hope for early detection and improved patient care, though further validation through large-scale clinical trials is necessary. With a better understanding of MPD and continued research efforts, we can hope to enhance the prognosis and quality of life for individuals affected by this condition.

## Figures and Tables

**Figure 1 cancers-15-05276-f001:**
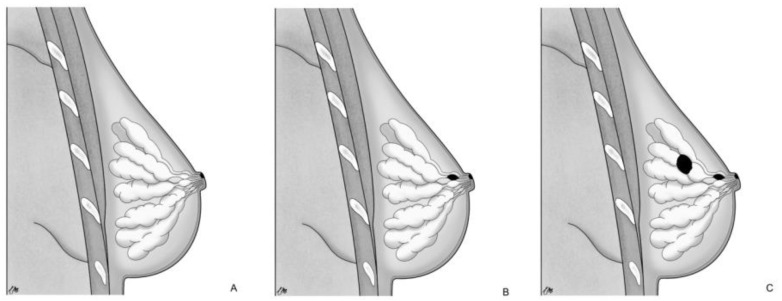
Pathogenetic classification of MPD: (**A**) only NAC-MPD without DCIS associated; (**B**) NAC-MPD with underlying lactiferous ducts DCIS; and (**C**) NAC-MPD with underlying lactiferous ducts DCIS and DCIS or invasive BC elsewhere at least 2 cm from the NAC.

**Figure 2 cancers-15-05276-f002:**
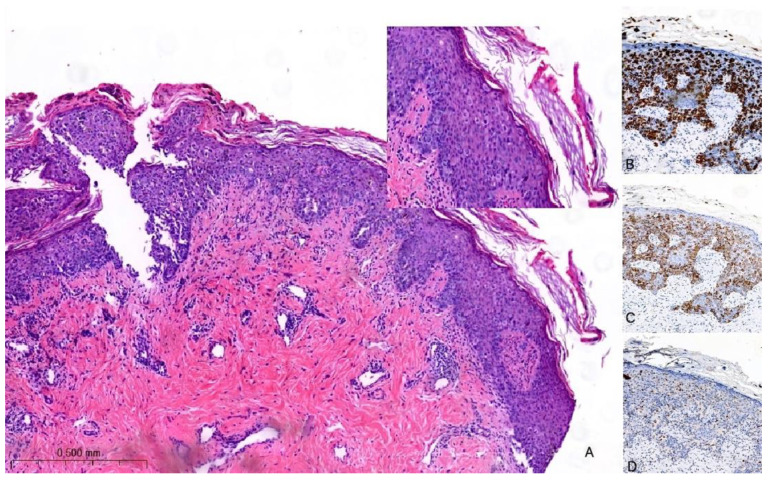
Hematoxylin and eosin stain of histological sections of MPD (**A**): large intraepidermal cells with abundant pale cytoplasm and pleomorphic nuclei containing prominent nucleoli (Paget cells, PC) (H-E 100×, 400×). Immunohistochemical staining of PC with (**B**) CK7 antibody (200×) shows a strong PC positivity; with (**C**) HER2 antibody (200×) shows positive staining (membrane); and with (**D**) SOX10 antibody (200×) is used for the differential diagnosis with melanoma: PC are negative.

**Figure 3 cancers-15-05276-f003:**
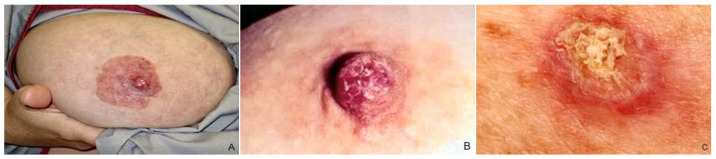
Clinical features of MPD: (**A**) NAC with significant erythema, scaling, and erosion; (**B**,**C**) erythematous scaly patch with oozing and crusting in the nipple.

**Figure 4 cancers-15-05276-f004:**
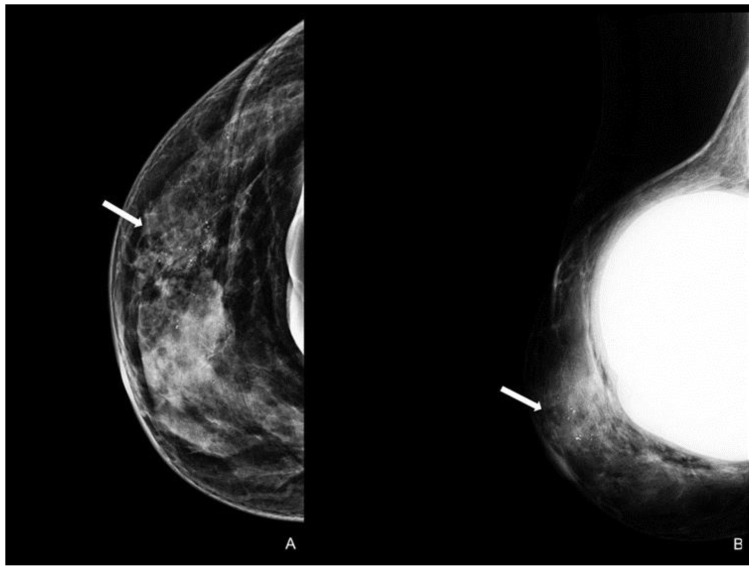
Woman with a bilateral breast augmentation and a history of recurrent eczema of the right nipple. A skin biopsy was performed, and MPD was confirmed. Unilateral craniocaudal (**A**) and oblique (**B**) mammograms of the right breast show periareolar skin thickening and diffuse fine pleomorphic microcalcifications (arrow). The patient underwent a stereotactic biopsy and was diagnosed with high-grade in situ ductal carcinoma.

**Figure 5 cancers-15-05276-f005:**
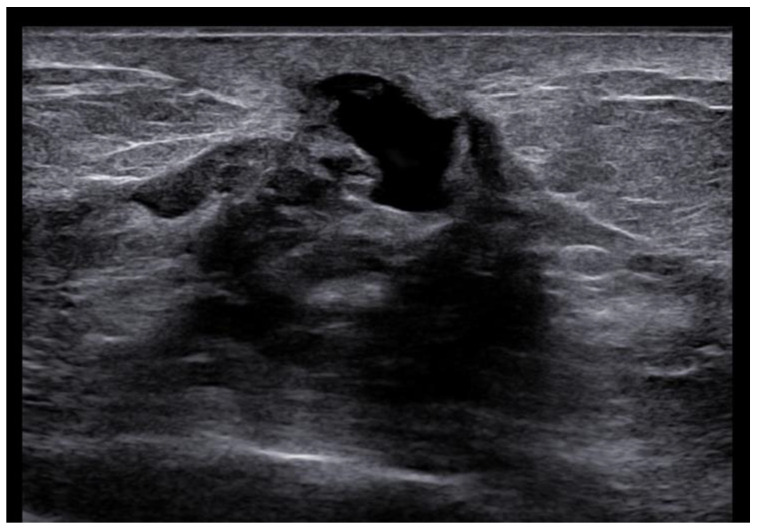
Woman with pruritic erythema and serous discharge of the right nipple and history of ipsilateral nipple-sparing mastectomy (10 years prior). Exfoliative cytology was performed with PC positivity. In the US, superficial hypoechogenic retroareolar areas with ectasia ductal are observed.

**Figure 6 cancers-15-05276-f006:**
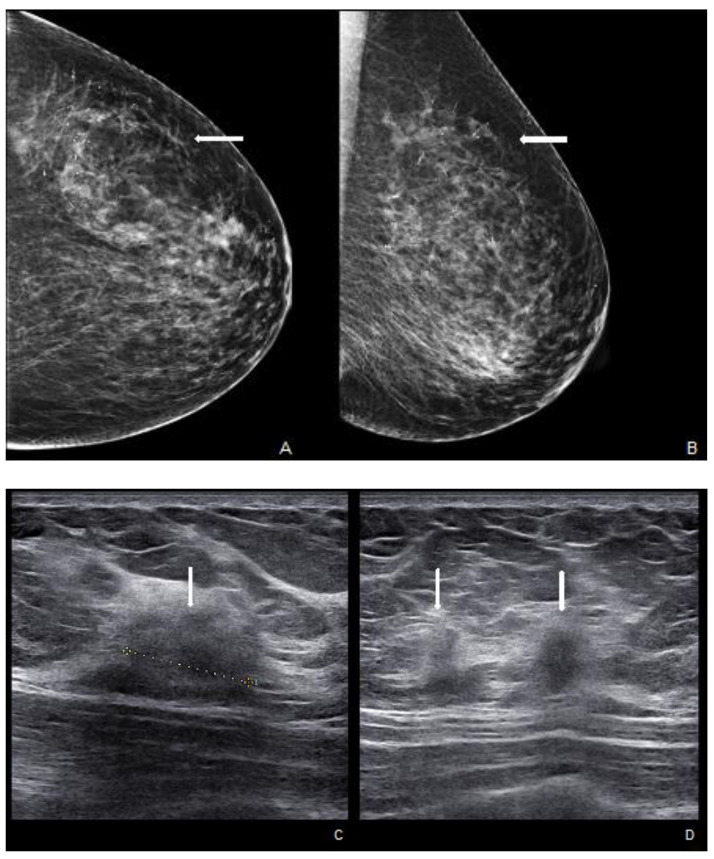
Woman with a palpable mass of the outer left quadrants and a history of pruritic erythema of the ipsilateral nipple. A skin biopsy was performed with PC positivity. Unilateral craniocaudal (**A**) and oblique (**B**) mammograms of the left breast show diffuse fine pleomorphic microcalcifications (arrow) in the upper outer quadrant with architectural distortion associated. US detects multiple inhomogeneous hypoechogenic areas (arrow) with poorly defined margins and maximum diameter of 15 mm (yellow line) (**C**,**D**), corresponding to foci of non-mass-like enhancement with associated nipple enhancement (**E**,**F**). The patient underwent a left mastectomy, and high-grade invasive e multicentric ductal carcinoma was diagnosed.

## Data Availability

Not applicable.

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
