# Peer review of "A Pictorial Exploration of Mammary Paget Disease: Insights and Perspectives"

_cancers, 2023, doi:10.3390/cancers15215276_

Round 1

Reviewer 1 Report

Comments and Suggestions for Authors

Author Response

Q1. On Page 6, Line 213, the authors mentioned that DM may appear negative in 22- 5% of patients. Do you mean that approximately 20-25% of patients have negative results on DM ?

Yes thank you for the suggestion. We have changed the percentage in the text.

Q2. In the Page 6, Line 225, the authors mentioned that nipple involvement even when it is clinical unsuspected for Paget’s disease. However, it is crucial to address the question of how to differentiate between tumor invasion of the nipple and Paget's disease using breast MRI.

We thank you for your comment, which allows us better to express the concept of MRI in PAGET pathology.

In the clinical suspicion of Paget's pathology, MRI can be very useful for two different but critical reasons:

- the evaluation of non-mass enhancement (not visible with other imaging methods), often associated with a co-existing in situ pathology.

- the evaluation of nipple infiltration by the pathological process.

We changed the text accordingly.

Q3. Figure 5: it may be necessary to include more specific images to clearly illustrate ductal ectasia and the precise location of the nipple.

Thank you for your suggestion. We have modified the image according to your advice.

Q4. Figure 6: the figures provided do not display any enhancement of the nipple.

Thank you for your suggestion. We have modified the image according to your advice.

Q5. It’s an interesting treatment algorithm mentioned by the authors, using the two conditions with NAC biopsy positive and imaging negative or positive. I am interested knowing which specific imaging modalities are included in this algorithm. Please clarify.

we thank you for the suggestion.

We have added a paragraph (p. 10 and 11 line 318-324) that better specifies the imaging modalities according to your request.

Q6. In Page 10, Line 337, the authors mentioned the integration of AI and Deep Learning models based on image analysis. Please discuss it.

we thank you for the suggestion.

We have added a paragraph (p. 12 line 390-407) that better discusses the role of artificial intelligence in breast imaging

Reviewer 2 Report

Comments and Suggestions for Authors

1. Part 7.1 --3, line 271--317,  can be combined in one paragraph. 

2. Part 9,    line 339--347,  the content of liquid biopsy has no relation to PDB, can be omitted.  

3. Line 350-- 352,  the content is inconsistent to Reference 66.

4. The format of references should be standardized.  

Author Response

  1. Part 7.1 --3, line 271--317,  can be combined in one paragraph. 

Thanks. We have modified the text according to your suggestion.

  1. Part 9,    line 339--347,  the content of liquid biopsy has no relation to PDB, can be omitted.  

Thanks. We have deleted the liquid biopsy paragraph in the text according to your suggestion.

  1. Line 350-- 352,  the content is inconsistent to Reference 66.

Thank you we apologize for the error. We have edited with the correct reference in the text.

(Alshammari FO, Satari AO, Aljabali AS, Al-Mahdy YS, Alabdallat YJ, Al-Sarayra YM, Alkhojah MA, Alwardat ARM, Haddad M, Al-Sarayreh SA, Al-Saraireh YM. Glypican-3 Differentiates Intraductal Carcinoma and Paget's Disease from Other Types of Breast Cancer. Medicina (Kaunas). 2022 Dec 30;59(1):86. doi: 10.3390/medicina59010086. PMID: 36676710; PMCID: PMC9862536.)

  1. The format of references should be standardized.  

             Thank you we have standardized the reference format according to the journal rules

Reviewer 3 Report

Comments and Suggestions for Authors

This is a comprehensive review of mammary Paget disease (MPD), covering its clinical, histologic, and radiologic aspects while also delving into treatment modalities. The manuscript is generally well-crafted, yet there are a few statements that require rectification. Firstly, both in the abstract and introduction, the authors assert that MPDs are "often" linked with underlying DCIS or IDC. However, this assertion is inaccurate. In contrast to its extramammary counterpart (EMPD), MPD is invariably associated with an underlying mammary carcinoma (either DCIS or IDC). Conversely, in EMPD, secondary EMPDs (i.e., Paget disease arising from an underlying internal malignancy, such as colorectal adenocarcinoma and urothelial carcinoma) are far less prevalent than primary EMPDs. To address this, it is suggested to amend "often" to "frequently" or "invariably". Another erroneous statement is the authors' assertion that MRI plays a pivotal role in MPD diagnosis. This is categorically untrue (and in fact, the authors themselves noted that histologic evaluation is essential to confirm the diagnosis of MPD). While MRI may aid in detecting the underlying mass-like lesion, it can NEVER confirm the diagnosis of MPD, rendering it irrelevant in this context. This should be corrected accordingly. The histopathological aspect of MPD lacks current information. A recent study introduced a novel immunohistochemical marker, TRPS1, which is now widely employed to affirm the diagnosis of MPD. According to this study, 100% of MPDs exhibited TRPS1 expression, aiding in distinguishing MPDs from histologic mimics, such as melanoma in situ. Please include an appropriate reference for this information [Cho et al. JCP 2023; PMID: 36808637]. Given that squamous cell carcinoma in situ almost never occurs in the nipple areolar complex, the diffuse expression of TRPS1 in an intraepithelial pagetoid lesion arising in the nipple invariably confirms the diagnosis of MPD. As the authors mentioned, CK7 and Cam5.2 are typically expressed in MPDs, but they can rarely be negative or only focally positive in MPDs and it is important to note that pagetoid squamous cell carcinoma in situ can also express these markers. Lastly, it is recommended to use the term "mammary Paget disease" (without an apostrophe after Paget) instead of "Paget's disease of the breast".

Author Response

This is a comprehensive review of mammary Paget disease (MPD), covering its clinical, histologic, and radiologic aspects while also delving into treatment modalities. The manuscript is generally well-crafted, yet there are a few statements that require rectification.

  1. Firstly, both in the abstract and introduction, the authors assert that MPDs are "often" linked with underlying DCIS or IDC. However, this assertion is inaccurate. In contrast to its extramammary counterpart (EMPD), MPD is invariably associated with an underlying mammary carcinoma (either DCIS or IDC). Conversely, in EMPD, secondary EMPDs (i.e., Paget disease arising from an underlying internal malignancy, such as colorectal adenocarcinoma and urothelial carcinoma) are far less prevalent than primary EMPDs.
    To address this, it is suggested to amend "often" to "frequently" or "invariably".

            Thank you for your comment and suggestion. We agree with you. we changed the text  according to your suggestions.

  1. Another erroneous statement is the authors' assertion that MRI plays a pivotal role in MPD diagnosis. This is categorically untrue (and in fact, the authors themselves noted that histologic evaluation is essential to confirm the diagnosis of MPD). While MRI may aid in detecting the underlying mass-like lesion, it can NEVER confirm the diagnosis of MPD, rendering it irrelevant in this context. This should be corrected accordingly.

Thank you for the suggestion. We agree with you.

We have changed the paragraph on resonance in the text. Pag 7 line 251-254

  1. The histopathological aspect of MPD lacks current information. A recent study introduced a novel immunohistochemical marker, TRPS1, which is now widely employed to affirm the diagnosis of MPD. According to this study, 100% of MPDs exhibited TRPS1 expression, aiding in distinguishing MPDs from histologic mimics, such as melanoma in situ. Please include an appropriate reference for this information [Cho et al. JCP 2023; PMID: 36808637]. Given that squamous cell carcinoma in situ almost never occurs in the nipple areolar complex, the diffuse expression of TRPS1 in an intraepithelial pagetoid lesion arising in the nipple invariably confirms the diagnosis of MPD. As the authors mentioned, CK7 and Cam5.2 are typically expressed in MPDs, but they can rarely be negative or only focally positive in MPDs and it is important to note that pagetoid squamous cell carcinoma in situ can also express these markers. Lastly, it is recommended to use the term "mammary Paget disease" (without an apostrophe after Paget) instead of "Paget's disease of the breast".

             Thank you for the suggestion.

              We have added and discussed the article you suggested (pag 4, line 156-164) and modified the term Paget disease according to your suggestion.

Round 2

Reviewer 3 Report

Comments and Suggestions for Authors

Thank you for incorporating my suggestions. I have no further comment.